# Quantifying Trade-Offs in the Choice of Ribosomal Barcoding Markers for Fungal Amplicon Sequencing: a Case Study on the Grapevine Trunk Mycobiome

Vinciane Monod,[a,b] Valérie Hofstetter,[b] Vivian Zufferey,[b] Olivier Viret,[c] Katia Gindro,[b] ⓘDaniel Croll[a]

[a]Laboratory of Evolutionary Genetics, Institute of Biology, University of Neuchâtel, Neuchâtel, Switzerland
[b]Agroscope, Plant Protection, Mycology, Nyon, Switzerland
[c]Direction générale de l'agriculture, de la viticulture et des affaires vétérinaires (DGAV), Département de l'Economie, de l'innovation et du sports (DEIS), Morges, Switzerland

**ABSTRACT**  The evolution of sequencing technology and multiplexing has rapidly expanded our ability to characterize fungal diversity in the environment. However, obtaining an unbiased assessment of the fungal community using ribosomal markers remains challenging. Longer amplicons were shown to improve taxonomic resolution and resolve ambiguities by reducing the risk of spurious operational taxonomic units. We examined the implications of barcoding strategies by amplifying and sequencing two ribosomal DNA fragments. We analyzed the performance of the full internal transcribed spacer (ITS) and a longer fragment including also a part of the 28S ribosomal subunit replicated on 60 grapevine trunk core samples. Grapevine trunks harbor highly diverse fungal communities with implications for disease development. Using identical handling, amplification, and sequencing procedures, we obtained higher sequencing depths for the shorter ITS amplicon. Despite the more limited access to polymorphism, the overall diversity in amplified sequence variants was higher for the shorter ITS amplicon. We detected no meaningful bias in the phylogenetic composition due to the amplicon choice across analyzed samples. Despite the increased resolution of the longer ITS-28S amplicon, the higher and more consistent yields of the shorter amplicons produced a clearer resolution of the fungal community of grapevine stem samples. Our study highlights that the choice of ribosomal amplicons should be carefully evaluated and adjusted according to specific goals.

**IMPORTANCE**  Surveying fungal communities is key to our understanding of ecological functions of diverse habitats. Fungal communities can inform about the resilience of agricultural ecosystems, risks to human health, and impacts of pathogens. Community compositions are typically analyzed using ribosomal DNA sequences. Due to technical limitations, most fungal community surveys were based on amplifying a short but highly variable fragment. Advances in sequencing technology enabled the use of longer fragments that can address some limitations of species identification. In this study, we examined the implications of choosing either a short or long ribosomal sequence fragment by replicating the analyses on 60 grapevine wood core samples. Using highly accurate long-read sequencing, we found that the shorter fragment produced substantially higher yields. The shorter fragment also revealed more sequence and species diversity. Our study highlights that the choice of ribosomal amplicons should be carefully evaluated and adjusted according to specific goals.

**KEYWORDS**  amplicon sequencing, grapevine, fungi, internal transcribed spacer, mycobiome, barcoding

Fungi occur in nearly all environments, are highly diverse, and can form tight associations with other organisms as pathogens or mutualists (1). The mycobiome associated with plants has important implications for agricultural ecosystems (2). Vascular diseases affecting

Address correspondence to Daniel Croll, daniel.croll@unine.ch.

The authors declare no conflict of interest.

plant stems, including xylem and phloem, are often difficult to diagnose or the causal agent is not yet known (3–6). Surveys of fungal communities (i.e., the mycobiome) have become key tools to understand how environmental and temporal factors influence species compositions associated with diseases (7, 8). The evolution of sequencing technology and multiplexing has rapidly expanded our ability to characterize fungal diversity in many environments (9). However, the implementation of molecular tools to establish unbiased mycobiome surveys remains challenging (9). Early impediments of surveying fungal diversity included the need to culture species, which creates significant biases in the estimation of community compositions (10, 11). Next-generation sequencing (NGS) technology has enabled the sequencing of taxonomically informative loci (i.e., barcoding) to reproducibly determine community structures and species diversity (1). Second-generation sequencing techniques (i.e., short-read sequencing) can generate deep-coverage amplicon data sets (12). Read length constraints limit amplicons to ca. 550 bp using an overlapping paired-end design (13).

Fungal nuclear ribosomal internal transcribed spacers 1 and 2 (ITS1 and ITS2), in addition to the 5.8S subunit, constitute the prevalent barcoding locus for fungi (14). With a typical amplicon length of 400 to 600 bp, the locus is compatible with second-generation sequencing read length limitations (15). However, several studies have highlighted potential shortcomings of relying on short barcoding markers (16, 17). Targeting longer amplicons has been made possible by third-generation sequencing technology. Longer amplicons were shown to improve taxonomic resolution (18) and resolve ambiguities by reducing the risk of spurious operational taxonomic units (OTUs) (19–21). An important factor in establishing long-read amplicon sequencing is error correction approaches such as PacBio circular consensus sequencing (CCS). CCS drastically reduces base-calling errors through multiple sequencing passes on the same molecules (19). Importantly, CCS provides a per-base accuracy comparable to that of short-read sequencing (22, 23). However, the choice of the locus, challenges in amplifying longer fragments, and downstream analyses remain important considerations.

Comparisons between long and short amplicon studies of fungal communities show that long amplicons have typically lower taxonomic coverage in sequence databases (24). However, longer reads can improve taxonomic resolution (25, 26). This raises the question of whether targeting a longer amplicon (i.e., full ITS or full ITS plus flanking rRNA subunit regions) can provide sufficient sequence read depth per sample to accurately capture relevant differences in species richness and community composition (27). Use of the full ITS region combines the benefits of capturing both ITS1 and ITS2 subregions (20). Targeting the full 5.8S rRNA gene provides improvements for fungal identification, notably the precision of genus-level identification because of a much lower substitution rate than for ITS1 or ITS2 (1).

The higher taxonomic resolution given by the full ITS can be used for strain-level identifications (28, 29). Tedersoo et al. (26) have shown that the identification rate was 33% higher at genus rank when using full-length ITS sequences than when using either ITS1 or ITS2. Longer sequences that combine the ITS with a portion of the small (18S) or large (28S) nuclear ribosomal subunit can also facilitate taxonomic assignments at the family or order level (17). In addition, more than 50% of taxonomically unassigned fungal ITS sequences, i.e., sequences corresponding to species belonging to underrepresented or not-yet-represented groups in sequence databases, can be identified at least at the divisional level by adding flanking ribosomal DNA (rDNA) regions (26). Assessing the impact of using one or more ribosomal DNA regions for fungal identification should inform decisions about the design of fungal community and barcode studies.

As a model to assess fungal barcoding amplicon suitability, we focused on the complex grapevine trunk mycobiome. The grapevine trunk is inhabited by various fungal species from different taxonomic and functional groups (30). Determining fungal diversity is of high interest because various dieback diseases are thought to be caused by fungal pathogens, constituting severe threats to vineyards worldwide with substantial economic consequences (31). Grapevine is subject to a complex set of interacting pathogenic or commensal microorganisms (2). Despite significant efforts over the past 3 decades, the causal agents among the grapevine trunk microbiome and the outbreak dynamics are poorly understood (30, 32). Expansive characterizations of the fungal diversity present in grapevine trunks may help to

**FIG 1** Trunk sample collection and barcoding loci. Shown are the sampling method used to extract wood cores from the grafting point of vine plants ($n = 60$) of a single vineyard before proceeding to DNA extraction, the genomic regions targeted for the amplification (internal transcribed spacer [ITS] and ITS-large subunit [LSU/28S] using the ITS1F-ITS4 and ITS1F-LR5 primer pairs, respectively), the bioinformatics workflow to filter and trim sequences following the DADA2 method, and inference of amplicon sequence variants (ASVs) from sequencing data and taxonomic assignments of each ASV.

identify one or more fungal species strongly associated with disease outbreaks. To date, the grapevine mycobiome has been analyzed largely based on culture-dependent approaches. Up to 159 OTUs were described using Sanger sequencing (31) and up to 259 OTUs with second-generation NGS technology (30). Third-generation long-fragment sequencing has not been used to describe the grapevine mycobiome to our knowledge.

In this study, we examined the implications of barcoding strategies in the context of the grapevine trunk mycobiome using third-generation sequencing technology. We amplified and sequenced two fragments to analyze their performance in characterizing the grapevine trunk mycobiome: the full ITS (14) and a longer fragment composed of the full ITS and a part of the large subunit (LSU) (ITS-28S) to assess the impact on taxonomic resolution and phylogenetic coverage. We analyzed whether the length of the target amplicon influences the sequencing depth and whether the sequencing depth correlates with the detected diversity. Finally, we examined if the amplicon choice influenced the detected fungal diversity.

## RESULTS

**Amplicon sequencing and read recovery.** A total of 60 grapevine plants located in a single plot were sampled at the grafting point using wood cores (Fig. 1). The prevalence of grapevine trunk diseases (GTD) among the 60 randomly selected plants was 14% in the sampling year. DNA extracted from the 60 wood samples could be successfully amplified in 59 and 50 samples with the primer pairs ITS1F-ITS4 (ITS) and ITS1F-LR5 (ITS-28S), respectively. One ITS and 10 ITS-28S amplicon samples were not sequenced because of a too-low PCR

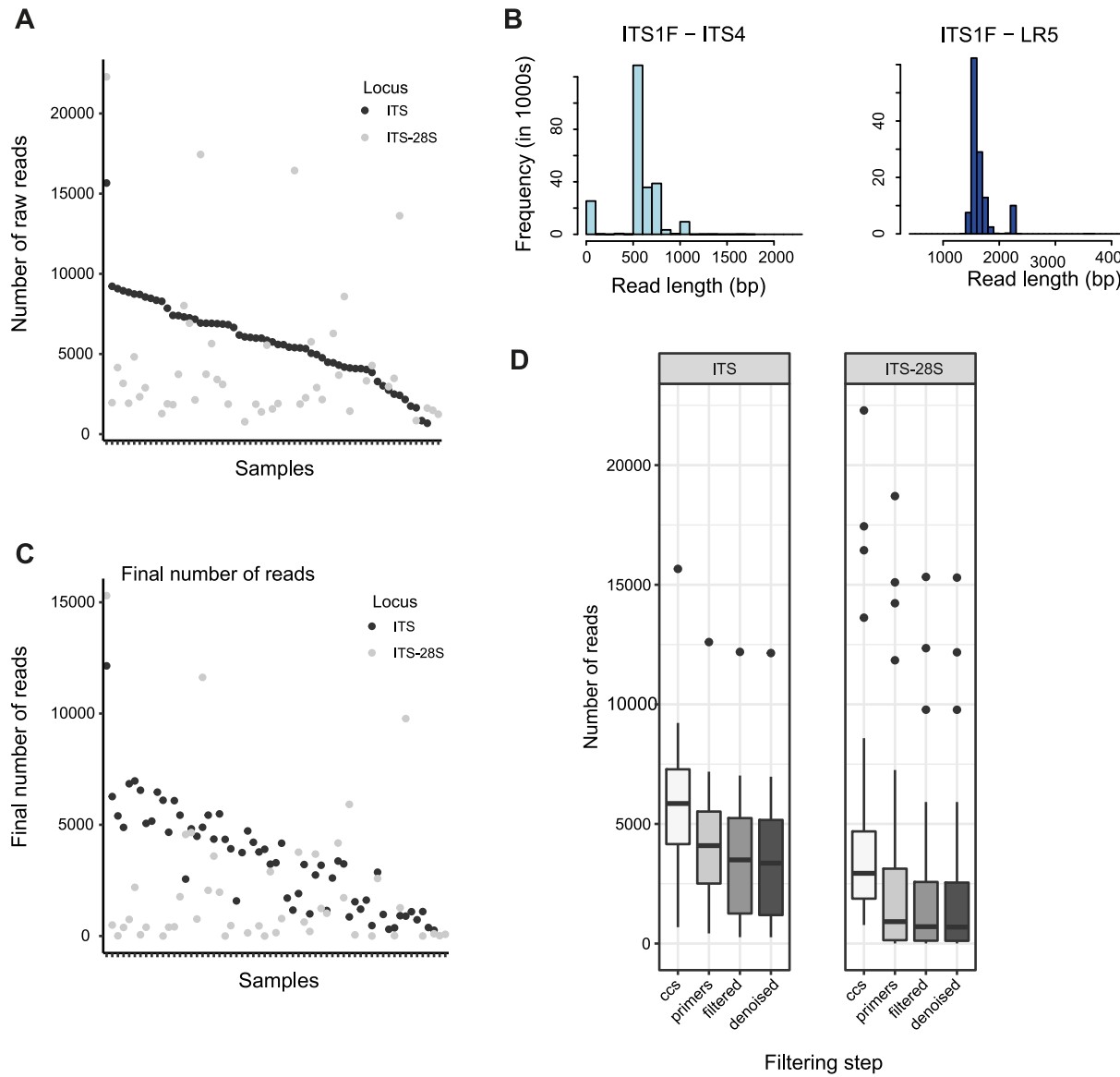

**FIG 2** Circular consensus sequencing (CCS) analyses of two ribosomal amplicons. (A) Number of raw CCS reads obtained for each of the 109 samples for the ITS and ITS-large subunit (28S) amplicons amplified using the ITS1F-ITS4 and ITS1F-LR5 primer pairs, respectively. Samples are ranked by raw read counts of the ITS amplicon. (B) Distribution of raw read lengths for each of the two amplicons for all samples combined. (C) Number of final read numbers for the two amplicons after all filtering steps. Samples are ranked by raw read counts of the ITS amplicon. (D) Impact of individual filtering steps (dereplication, primer detection, filtering for amplicon length, denoising according to error detection, and presence of chimeras) on the read counts per sample for each of the two amplicons.

product yield after amplification. A total of 382,672 PacBio CCS reads were successfully demultiplexed using the 8-bp barcode with 100% identity, generating 682 to 15,663 reads per sample for the ITS (>4,156 reads for 75% of the samples). For the ITS-28S, 227,549 raw reads were demultiplexed, generating 707 to 22,288 reads per samples (>1,876 reads for 75% of the samples). Sequencing depths were highly variable among samples for both loci (Fig. 2A). A total of 90% of the reads were successfully demultiplexed for the ITS and 89% for the ITS-28S. In proportion, the ITS locus accounted for 63% of the total number of reads. The number of reads per sample between the two amplicons was not correlated (Fig. 2A). This shows that the equimolar pooling of the PCR products was successful in balancing sequencing yields. Sequence lengths were comparable to the expected PCR amplicon sizes. The average sequence lengths were 629 bp for the ITS and 1,526 bp for the ITS-28S (Fig. 2B).

Dereplication according to detected flanking primer sequences is a critical step to ensure high-quality sequences, but this step can also discard a substantial number of

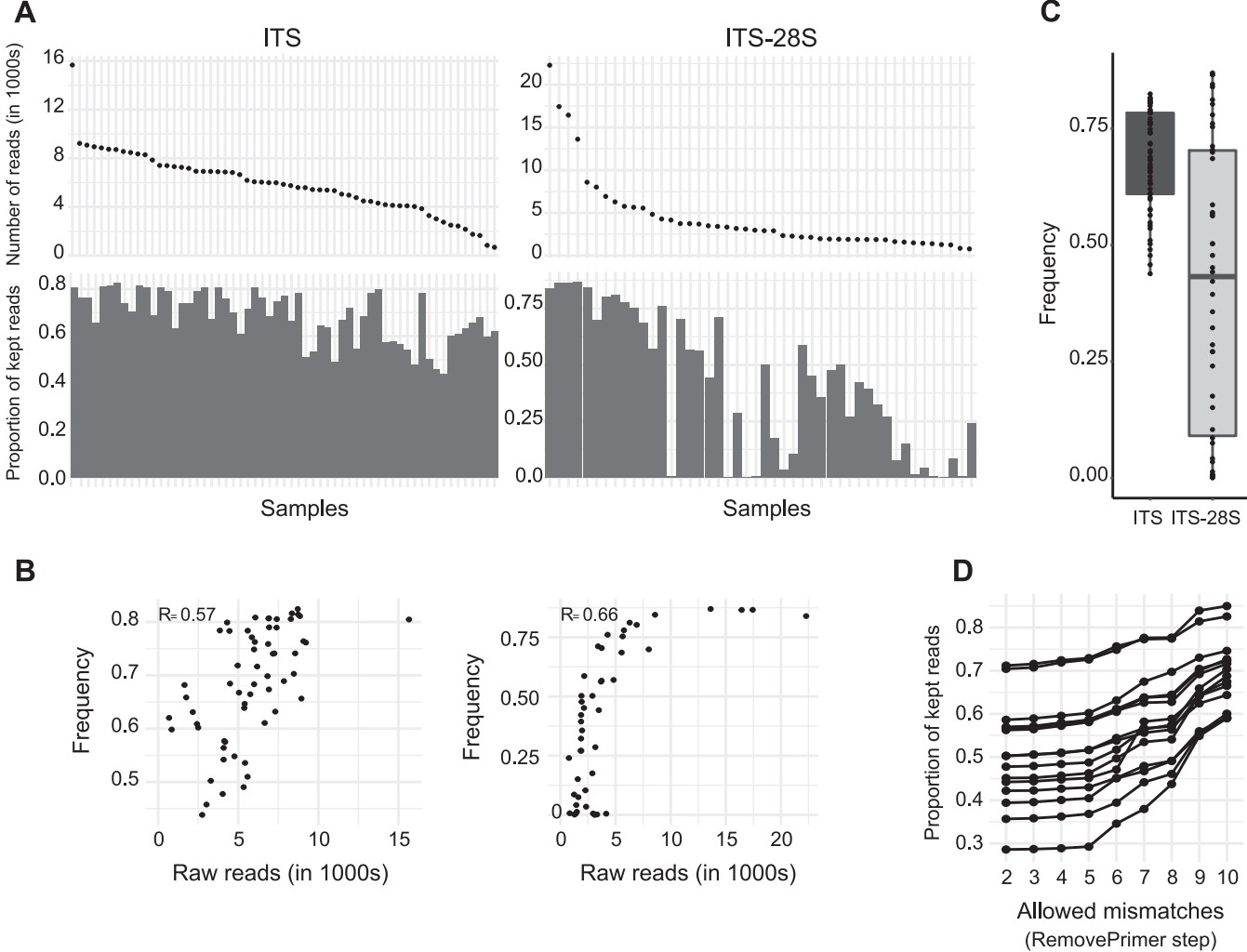

**FIG 3** Impact of read filtering steps. (A) Samples ranked by raw read counts and proportion of kept reads by sample at the primer trimming step for the ITS and ITS-28S. (B) Relation between raw read counts and proportion of kept reads at the primer trimming step for the ITS and ITS-28S. (C) Proportion of retained reads at the primer trimming step for both markers. (D) Proportion of kept reads according to the number of allowed mismatches in primers detection for ITS-28S.

sequences. For the ITS, 29% of reads were discarded at this step; 38% were discarded for the ITS-28S. Three samples from the ITS-28S saw no reads passing the dereplication, as no matching primer sequences were detected. Read filtering for amplicon length showed disparities between the ITS and ITS-28S data sets, with 70% of reads retained for the ITS and 61% retained for the ITS-28S (Fig. 2D). Error detection and denoising retained most of the reads for both loci ($\geq$98%). Chimeras were identified in $\sim$2% of the ITS sequence reads but in <1% for ITS-28S sequence reads. At the end of the quality filtering procedure, 58% of the reads were retained for the ITS and 45% for the ITS-28S. The final sequencing depth was 1,190 reads or more for 75% of the samples for the ITS, but it was only 112 reads or more for 75% of the samples for the ITS-28S. Hence, the ITS amplicon produced significantly more high-quality reads after the filtering steps. The two target amplicons differed in total sequencing yield and proportion of reads kept after the quality filtering steps (Fig. 2C and D). Successfully identifying both primer sequences was a critical step for read retention during the ITS-28S filtering process. The ITS-28S amplicons showed also more variability among samples in retained read proportions than the ITS (Fig. 3A to C). Samples with more reads showed a weak tendency to have a higher proportion of kept reads ($R = 0.57$ [Fig. 3B]). This tendency was stronger for ITS-28S amplicons ($R = 0.66$). To better understand the shifts between the two data sets, we tested multiple filter parameter values for their impact on read retention. However, more relaxed primer matching parameters did not meaningfully

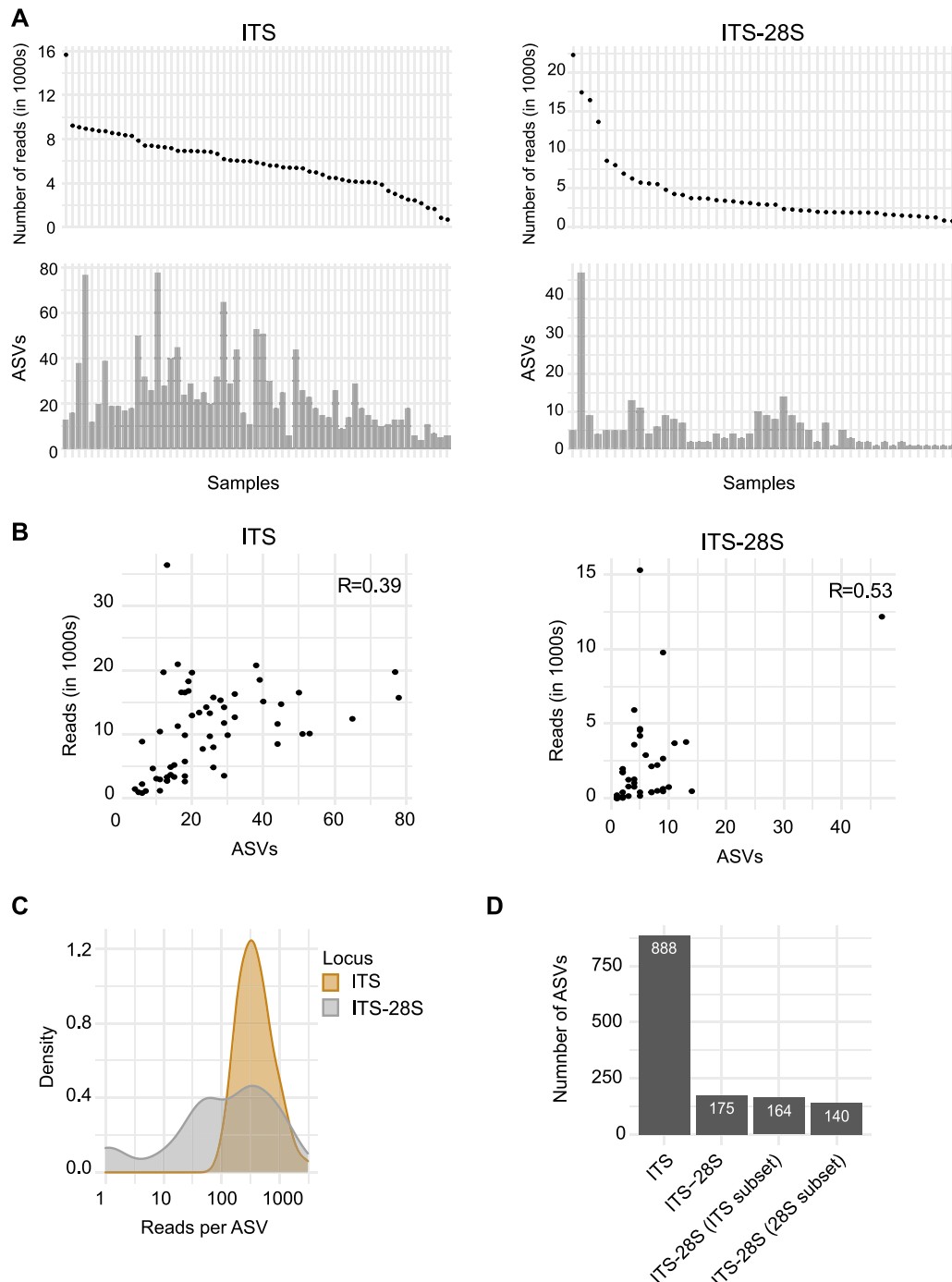

**FIG 4** ASV diversity. (A) Samples ranked by raw read counts and inferred ASVs by sample for the ITS and ITS-28S. (B) Relation between raw read counts and ASVs detected for the ITS and ITS-28S. (C) Distribution of reads by ASVs for the ITS and ITS-28S. (D) ASVs detected by markers.

increase retained reads. This is true for up to six allowed mismatches for primer detection. Beyond six allowed mismatches, a higher proportion of reads were kept as expected (Fig. 3D). In summary, the difference in retained reads between amplicon data sets is largely unrelated to the primer matching stringency.

**Comparison of sequence variant recovery for the two barcoding loci.** The number of inferred amplified sequence variants (ASVs) was highly variable among samples for both loci (Fig. 4A). We found only a moderate correlation between the number of reads and the ASVs inferred per sample ($R = 0.39$ for the ITS and $R = 0.53$ for the ITS-28S [Fig. 4B]). For

the ITS, we found a strong density peak of 500 reads by unique ASVs with a normal distribution. For the ITS-28S data set, the distribution of reads by ASVs was more variable, with a flatter curve going from 50 to 1,000 reads by unique ASVs (Fig. 4C). Overall, 933 ASVs were detected including both data sets. The ITS data set comprises 888 ASVs among 59 samples; the ITS-28S data set comprises 175 ASVs among 46 samples. The artificially cut ITS-28S fragment still covered 164 ASVs for the ITS subset and 140 for the 28S subset (Fig. 4D). The subset creation for the long fragment provides a direct comparison of the represented sequences in the ITS and ITS-28S amplicon data sets. In a direct comparison, we found 888 ASVs with the ITS and 164 ASVs with the ITS-28S amplicon subset to the ITS, of which 119 (12.7%) were shared among the two amplicon data sets (Fig. 5A). A total of 45 ASVs were detected only by the ITS-28S subset to the ITS amplicon and 769 ASVs were detected only by the ITS amplicon. Among the shared ASVs, the proportions occupied by individual ASVs in the two data sets were very similar. Overall, 83% of the shared ASVs differed by less than 1% in relative proportion between the two amplicon sets (Fig. 5B). The most differentiated ASVs in terms of relative proportion were ASVs assigned to *Fomitiporia punctata* (8% more abundant in the ITS-28S subset to ITS) and *Bacidina neosquamulosa* (4% more present in the ITS). Hence, the long fragment revealed only a minor degree of additional ASVs not already captured by the ITS amplicon.

**Community composition of the grapevine mycobiome.** To better understand the consequences of targeting either the ITS or the ITS-28S on the detected fungal community composition, we examined the diversity present in both data sets across several taxonomic ranks. First, we created a subset of the ITS-28S amplicon consisting of a more conserved LSU portion. This subset of the ITS-28S typically provided only genus-level resolution using the RDP database. For the ITS portion of the ITS-28S amplicon, as well as the ITS amplicon, ≥60% of ASVs were assigned at the species level (Fig. 5C). Next, we analyzed the relative abundance of reads assigned to each phylum. Ascomycota represented the highest proportion of the reads (77%) in the ITS data set and slightly less in the ITS-28S ITS subset (67%) and 28S subset (65%). Basidiomycota proportions showed opposite patterns, with the ITS data set showing 22%, the ITS-28S ITS subset showing 33%, and the 28S subset showing 33% (Fig. 5E). For classes represented by more than 2% of the reads, we identified Eurotiomycetes and Agaricomycetes as the most abundant in all data sets. Dothideomycetes, Sordariomycetes, and Lecanoromycetes were represented by >2% of the reads only in the ITS data set. Similarly, Exobasidiomycetes were only found in the ITS-28S 28S subset at ≥2% (Fig. 5E). The most represented genera according to the ITS amplicon were *Phaeomoniella* (40%), *Fomitiporia* (9%), and *Bacidina* (4.6%). For the ITS-28S ITS subset, the genera were similarly *Phaeomoniella* (46%), *Fomitiporia* (15%), and *Mollisia* (7%). For the ITS-28S 28S subset, we found *Xenocylindrosporium* (25%), *Fomitiporia* (16%), and *Mollisia* (8%) (Fig. 5E).

At the species level, the diversity detected for the ITS-28S ITS subset was 45 ASVs. These ASVs correspond to 17 taxa with around half ($n = 8$) of the species or genera unable to be detected with the ITS marker (see Table S1 in the supplemental material). The taxa detected only by the ITS subset of the ITS-28S amplicon were *Mollisia* (8,301 sequences), *Pseudoophiobolus rosae* (49 sequences), *Meyerozyma guilliermondii* (19 sequences), *Coniochaeta coluteae* (9 sequences), *Bipolaris drechsleri* (6 sequences), *Seimatosporium pistaciae* (6 sequences), *Wojnowiciella cissampeli* (3 sequences), and the Ceratobasidiaceae family (2 sequences). For the 769 ASVs detected only based on the ITS amplicon, 103 species corresponded to species uniquely detected by the ITS amplicon. The most abundant species in the data set typically included multiple distinct ASVs matching to the same species. A total of 50% (ITS and ITS-28S 28S subset) and 60% (ITS-28S ITS subset) of the detected genera were represented by unique ASVs. Around 18% (ITS and ITS-28S ITS subset) and 16% (ITS-28S 28S subset) of the genera were represented by three distinct ASVs (Fig. 5D).

The species represented by the highest number of ASVs was *Phaeomoniella chlamydospora* for both data sets. This species was represented by 304 (ITS) and 75 (ITS-28S ITS subsets) different ASVs, highlighting significant intraspecific variation. *P. chlamydospora* was also the most abundant species in the ITS and ITS-28S ITS subset data sets. This species, which is typically associated with grapevine trunk disease, was identified on 88% (ITS) and 97% (ITS-28S ITS subset) of the sampled plant. *P. chlamydospora* was not detected in the ITS-28S 28S

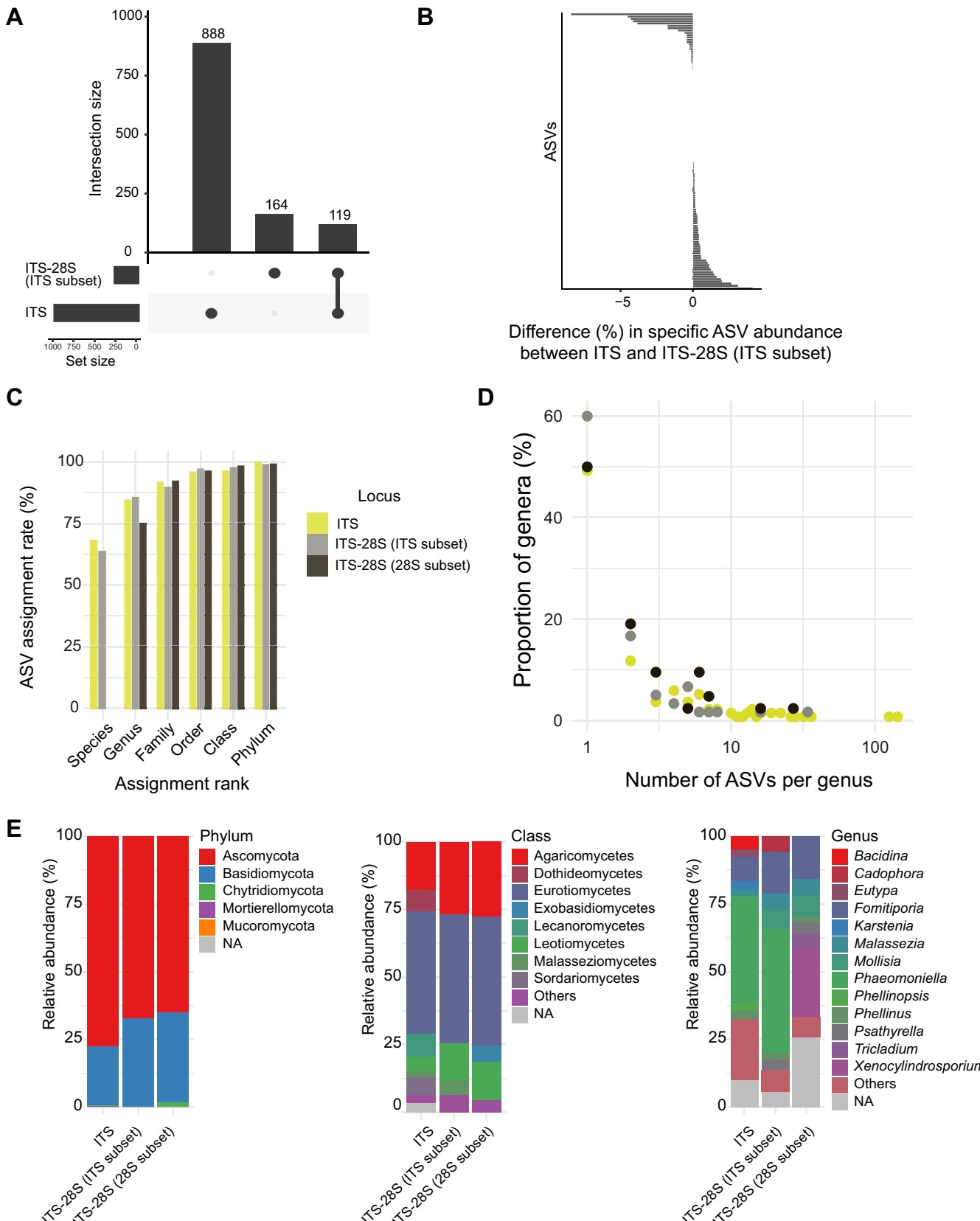

**FIG 5** Taxonomic diversity among grapevine trunk samples. (A) Intersecting sets of ASVs from ITS and ITS-28S ITS subset data set. (B) Proportional divergence of the shared ASVs of ITS and ITS-28S ITS subset. (C) Proportion of assigned reads at several taxonomic ranks. (D) Proportion of genera by number of ASVs for ITS, ITS-28S ITS subset and ITS-28S 28S subset. (E) Relative abundance of several taxonomic ranks (phylum, class, and genus) for the ITS, ITS-28S ITS subset, and ITS-28S 28S subset.

subset. As this was surprising, we independently analyzed assignments of some ASVs using BLAST searches in NCBI GenBank. Some ASVs classified as belonging to the *Xenocylindrosporium* genus on the basis of matches in the RDP database rather belong to the *Phaeomoniella* genus based on GenBank matches. Uncertainty about taxonomic assignments using RDP is consistent with concerns about representativeness issues of long ribosomal fragment databases.

**OTU-based clustering and contrast to ASV analyses.** We used the same error-corrected sequences as for the ASV-based analyses to obtain operational taxonomic units (OTUs) prior to taxonomic assignments. Using the commonly used threshold of 97% sequence identity, we obtained 559 OTUs for the ITS fragment (versus 888 ASVs), 102 OTUs for the ITS-28S subset to the ITS (versus 164 ASVs), and 70 OTUs for the LSU-28S subset to the LSU (versus 140 ASVs). We examined the proportion of genera assigned to each OTU for each marker set. Overall, the proportions of assigned genera differed very little, with typically <1% changes in the assigned categories, including the proportion of unassigned sequences (Table S2). Using the 99% threshold for OTU assignments, we observed a spike in the proportion of OTUs without assigned genera compared to other OTU thresholds and ASV-based assignments.

## DISCUSSION

We used PacBio long-read technology to amplify and sequence two fungal barcoding amplicons (ITS and ITS-28S) in parallel from DNA extracted from a set of vine wood samples. Using identical handling, amplification, and sequencing procedures, we obtained higher sequencing depth and higher ASV diversity for the shorter amplicon (i.e., ITS). We found no meaningful bias in the phylogenetic representation of the samples according to the selected amplicon. Despite the increased resolution of the long ITS-28S amplicon, the higher and more consistent yields of the shorter amplicons produced a clearer resolution of the fungal community of grapevine stems.

**Recovery of fungal barcoding sequences.** The PacBio sequencing libraries prepared in parallel for the two different ribosomal amplicons differed in yield, with ~1.5 times more sequences obtained for the ITS. Similarly, quality filtering retained a higher proportion of sequencing reads from the shorter amplicon. The higher yield and quality were consistent with findings by Tedersoo et al. (26). Differences in the number of sequences recovered for the two amplicons could be due to less efficient PCR amplification, e.g., due to competition among primed amplicons during the adaptor ligation step (26). Another possible explanation is that LR5 might be less efficient than ITS4 to amplify fungal diversity (33). Longer amplicons can also show reduced yields due to template positioning in sequencing wells (24, 26). Difficulties in properly detecting primer sequences on the long fragment could be due to a deterioration of sequencing quality. However, circular consensus sequencing should produce homogeneous quality scores for the entire template. Technological progress with the Sequel I system of PacBio and library preparation overall likely benefited read retention during quality filtering. Both of our amplicons showed higher retention (58% for ITS and 45% for ITS-28S) than obtained by Tedersoo et al. (26) (28% of retained reads with a Sequel I system for a 400- to 700-bp amplicon and 24% with the RSII system for a 1,250- to 1,700-bp amplicon). With further progress, sequencing of longer fragments should become even more efficient.

**Contrasting the recovered sequence diversity.** Targeting either a shorter amplicon with higher sequencing depth or a longer amplicon with reduced depth creates a trade-off that needs to be resolved. Both longer fragments and higher depth have the potential to improve the resolution of taxa present in a sample. We detected more ASVs using the ITS data set, but we found no clear relationship between the sequencing depth and detected diversity as expected. As our study covered environmental samples, the sequence diversity is most likely highly heterogeneous. Hence, sequencing depth versus sequence diversity correlations among samples are only of limited use. Indeed, we detected samples of intermediate sequencing depth but with some of the highest numbers of recovered ASVs. Above some threshold, increasing sequencing depth is not expected to yield more recovered diversity (27). Purahong et al. (21) showed also that sequencing depth alone is a poor performance metric to evaluate the representation of the fungal community. Kennedy et al. (27) suggested a threshold of 100 reads per sample, beyond which additional reads are unlikely to substantially shift community assessment of environmental samples. In our study, we obtained

>1,190 reads for the ITS and >112 reads for the ITS-28S amplicons for 75% of the samples. Hence, regardless of the selected amplicon, fungal communities should be reasonably well assessed in our study.

A basic argument for preferring longer amplicon sequences is the ability to detect a larger number of sequence variants present in the data sets (i.e., ASVs). Interestingly, our study recovered only a few ASVs using the longer ITS-28S that were not detected by the shorter amplicon (45 ASVs). It is likely that we have somewhat underestimated ASV diversity based on the shorter amplicon due to the chosen length cutoffs. Hence, using ITS is more beneficial to assess the fungal diversity present in the analyzed grapevine trunk samples than targeting a longer amplicon making use of the highly accurate PacBio consensus reads. Some studies have even reported that increasing target amplicon lengths has negative effects on the assessment of microbial richness and community composition (34, 35). Our own comparative analyses have not revealed such detrimental effects. ASVs recovered with the ITS and ITS-28S amplicons showed very similar relative abundances.

**Recovery of taxonomic diversity.** We compared how well amplicons could be assigned to different taxonomic levels depending on the marker targeted. Taxonomic identification for the ITS data set was similar to that for the ITS sequence extracted from the longer amplicon as well as for the entire ITS-28S amplicon (>75% of the ASVs assigned to the genus level for the three amplicon data sets). A somewhat lower proportion of the 28S sequences extracted from the ITS-28S amplicon were assigned, which is most likely explained by the poor taxon coverage of the 28S subunit, compared to ITS, in sequence databases (~15 times more ITS sequences available in UNITE). The generally high taxonomic assignment rates are consistent with the general growth of amplicon databases and the high accuracy of consensus sequences compared to the case with previous studies relying on short-read sequencing (i.e., Illumina). Furthermore, improvements in the analysis pipelines (i.e., DADA2) reduced erroneous chimera sequences and increased the accuracy of amplicon data sets. Our parallel diversity analyses based on two ribosomal amplicons revealed highly diverse fungal communities across grapevine trunks sampled across a vineyard. Our findings are consistent with previous analyses by Del Frari et al. (30) and Travadon et al. (36). We found high consistency in the identified taxa (8 out of 14 genera identified by Del Frari [30]). The consistently recovered taxa include *Phaeomoniella chlamydospora*, *Phaeoacremonium* spp., and *Fomitiporia mediterranea*, fungal species commonly associated with esca disease of grapevine (37). This is consistent with expectations for diversity of fungi sampled at the grafting point. Our implementation of PacBio amplicon sequencing opens up opportunities for high-resolution profiling across large sets of samples covering space and time in mature vineyards. The high precision of the recovered sequences will allow the monitoring of fungal strains associated with GTD and more generally of wood endophyte fungal species.

Limitations in our comparison lie in the lack of mock community analyses. Comparing the resolutions of barcoding markers on reference DNA mixtures but also more realistic mock analyses of wood core extracts will further substantiate our performance assessments of ribosomal profiling. Additionally, a truly comparable reference database for taxonomic assignments would be needed. Excising the ITS portion of the longer amplicon allowed for a direct comparison using the same database, but a full comparison between the shorter and longer amplicon would require a fully equivalent ITS-28S sequence database. Our analyses match findings by Brown et al. (38) and Porras-Alfaro et al. (39) similarly comparing ITS and 28S amplicon diversity. It is evident that progress in exploiting longer amplicons to their full potential will require more comprehensive databases commensurate with the progress of sequencing technology.

## MATERIALS AND METHODS

**Sample collection.** Grapevine trunk sampling was conducted in La Côte vineyards in Echichens, Switzerland (46°32′07.034″N, 6°30′03.807″E, WGS 84, 475 m above sea). Samples were collected from 60 different plants in a single vineyard plot of 150 m by 40 m. Grapevine plants were all of the variety Gamaret (a cross between Gamay and Reichensteiner) grafted onto 3309C rootstock (*Vitis riparia* × *Vitis rupestris*). All the plants came from the same nursery (Dutruy in Founex, Switzerland) and were planted in 2003. Plants affected by GTD had been replaced continuously and were not considered in our sampling. A plant was considered diseased if one or more shoots showed leaf necrosis up to wilt or dieback symptoms. These are typically symptoms

classified as grapevine trunk disease (4, 40, 41). Grapevine wood was sampled at the grafting point (where fungal diversity was previously shown to be highest) using a nondestructive method. A 0.5-cm² piece of bark was removed with a surface-sterilized (80% ethyl alcohol [EtOH]) scalpel. The sampling was then performed with a power drill with a surface-sterilized drill bit (Ø 3.5 mm) by running the drill gently where the bark was removed to collect the coiled wood (∼60 mg) in an Eppendorf tube held underneath with sterilized tweezers. In the Eppendorf tubes, kept in an ice box during the sampling process, two 5-mm iron beads had previously been deposited to ease the next step of the protocol. As soon as possible, the Eppendorf tubes containing the coiled wood were stored at −80°C (Fig. 1).

**DNA extraction from wood samples.** Eppendorf tubes (safe lock) containing two 5-mm iron beads and wood samples were taken out of the −80°C freezer and put in liquid nitrogen. The Eppendorf tubes were then placed two times for 1 min at 30 Hz in a TissueLyser (Qiagen Inc., Germantown, MD, USA) to disrupt wood tissues. Between and after these two steps of tissue disruption, tubes were placed in liquid nitrogen for 1 min. After the tubes were placed on ice to let them gently thaw, 1 mL of cetyltrimethylammonium bromide (CTAB) was poured into each tube. The samples were then centrifuged for 1 min at 15,000 rounds/min and the supernatant was transferred to a new tube. The fungal DNA extraction was then performed with phenol-chloroform as described by Hofstetter et al. (42). The DNA quality was checked with an electrophoresis gel, and the extracted products were stored at −80°C.

**Amplification of fungal ribosomal DNA.** Two loci were targeted for amplification: ITS using primers ITS1F (CTTGGTCATTTAGAGGAAGTAA) and ITS4 (TCCTCCGCTTATTGATATGC) and a longer amplicon including ITS and a portion of the 28S subunit using primers ITS1F and LR5 (TCCTGAGGGAAACTTCG) (Fig. 1). We followed the PacBio procedure using barcoded universal primers for multiplexing amplicons, which includes two PCR steps (see https://www.pacb.com). The first PCR program was 30 s of denaturation at 98°C and then 30 cycles of 15 s at 98°C, 15 s at 55°C, and 1 min 30 s at 72°C, followed by a final elongation step for 7 min at 72°C. The second PCR program was 30 s of denaturation at 98°C and then 20 cycles of 15 s at 98°C, 15 s at 64°C, and 1 min 20 s at 72°C, followed by a final elongation step for 7 min at 72°C. We performed purification between the two PCRs to reduce contaminants or carryover of primer dimers using 96-well PCR purification plates (Qiagen Inc., Germantown, MD, USA). The final libraries were quantified with a Qubit fluorometer (Thermo Fisher, Foster City, CA, USA), and then all samples were pooled equimolarly. The pooled samples were then purified with 1× AMPure XP beads (Beckman Coulter Inc., Indianapolis, IN, USA) as per the manufacturer's instructions. Amplicons were prepared for SMRT sequencing at the Functional Genomics Center in Zürich (FGCZ), Switzerland. Sequencing was performed on the PacBio Sequel II platform.

**Demultiplexing and trimming.** The raw reads were demultiplexed using lima (https://lima.how/). After obtaining fastq files, reads were processed for quality filtering with the DADA2 package for R (43) (Fig. 1) (https://github.com/benjjneb/dada2). The DADA2 processing steps include the following: (i) dereplication with primer detection (reads without primer sequences are discarded); (ii) filtering for amplicon length (between 500 and 1,000 bp for the ITS and 1,000 to 2,000 bp for the longer fragment); (iii) error detection, in which error rates are learned by alternating between sample inference and error rate estimation until convergence (a feature table of observed transitions for each type and quality scores are produced); (iv) denoising to reduce sequencing errors based on error models; and (v) checking for chimeras, in which each sequence is evaluated against a set of putative parental sequences drawn from the sequence collection. Several tests were performed with removeBimeraDenovo by increasing the minFoldParentOverAbundance parameter from 4 to 8. We chose to continue with a minFoldParentOverAbundance of 8 to retain a maximum of reads.

**Analyses of amplicon sequence variants and taxonomic assignments.** We used the DADA2 algorithm to infer amplicon sequence variants (ASVs) from the filtered reads (Fig. 1). ASVs represent reads with 100% similarity accounting for sequencing errors by appropriately modeling PacBio CCS sequencing errors (18). Taxonomic assignments were performed with the function AssignTaxonomy of the DADA2 pipeline, which classifies sequences based on reference training data sets. The databases used for assignments were UNITE for the ITS (44, 45) and the Ribosomal Database Project (46) (RDP LSU training set). To compare sequence diversities directly between the short and long amplicon sets, ITS-28S fragments were cut in two subset fragments corresponding to the ITS and the 28S subunit (beginning of LSU), respectively. This was performed using the removePrimers step in the DADA2 pipeline. Instead of the ITS1F-LR5 primer pair as described above, we used ITS1F-ITS4 for the ITS and rcITS4-LR5 for the 28S subunit. The sequence subset creation produced the two subsets, ITS-28S subset to ITS and ITS-28S subset to 28S. OTU clustering was performed with the R package DECIPHER using three identity levels (95, 97, and 99% identity) in R (v4.1.2) (47, 48).

**Data availability.** All PacBio sequencing data are available from the NCBI Sequence Read Archive (SRA) under BioProject PRJNA847708.

## SUPPLEMENTAL MATERIAL

Supplemental material is available online only.
**SUPPLEMENTAL FILE 1**, XLSX file, 0.2 MB.

## ACKNOWLEDGMENTS

Library preparation and sequencing were performed at the Functional Genomics Centre Zurich (FGCZ). We are grateful to the members of the mycology research group of Agroscope (N. Lecoultre, E. Michellod, A.-L. Fabre, P.-H. Dubuis, D. Restori, and A. Melgar) for their assistance in sampling vineyards.

This work was funded by a research grant from the Canton de Vaud to K.G.

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
