## [Reviewer comments · Microbiology Spectrum]

Microbiology Spectrum

Quantifying trade-offs in the choice of ribosomal barcoding markers for fungal amplicon sequencing: a case study on the grapevine trunk mycobiome

Vinciane Monod, Valérie Hofstetter, Vivian Zufferey, Olivier Viret, Katia Gindro, and Daniel Croll

Corresponding Author(s): Daniel Croll, University of Neuchâtel

Review Timeline:

Submission Date:	July 11, 2022
Editorial Decision:	September 3, 2022
Revision Received:	November 1, 2022
Accepted:	November 5, 2022

Editor: Matthew Anderson

Reviewer(s): Disclosure of reviewer identity is with reference to reviewer comments included in decision letter(s). The following individuals involved in review of your submission have agreed to reveal their identity: Robert Lücking (Reviewer #1); Shwetha M Acharya (Reviewer #2)

Transaction Report:

DOI: <https://doi.org/10.1128/spectrum.02513-22>

September 3, 2022

Prof. Daniel Croll
University of Neuchâtel
Emile-Argand 11
Neuchâtel
Switzerland

Re: Spectrum02513-22 (Quantifying trade-offs in the choice of ribosomal barcoding markers for fungal amplicon sequencing: a case study on the grapevine trunk mycobiome)

Dear Prof. Daniel Croll:

The reviewers found the manuscript submitted by your team to be a strong body of work that was easy to read and follow. There are a few minor revisions that are requested here. These are not expected to take very long to address as these primarily deal with text edits and not additional analysis.

Link Not Available

Sincerely,

Matthew Anderson

Journals Department
Reviewer comments:

Reviewer #1 (Comments for the Author):

This study compares the differences in taxon detection and identification using different read lengths in a PacBio metabarcoding approach. Overall the study has been carefully elaborated and the manuscript is well written, and I cannot find any major flaws. However, I suggest the authors use the opportunity to add 97% OTU clustering to the methods, to enable comparison with their ASV approach when recovering taxon diversity. The ASV approach is more accurate, and so it would provide useful comparative data to have the taxonomic diversity reported for the same underlying data when doing 97% OTU clustering (overestimation?),

an outdated but still widely used approach.

Reviewer #2 (Comments for the Author):

The study illustrates the use of third-generation long read sequencing to characterize grapevine mycobiome. Authors test influence of different amplicon barcoding strategies for better taxonomic resolution and phylogenetic coverage. Based on the data shorter amplicon reads (full length ITS) has better taxonomic resolution and sequencing depth than longer ITS-28S amplicon reads. The manuscript is clear, concise and well-written.

There are a few minor suggestions/questions that need to be addressed:

1. Though not the main aim of the manuscript, it would be nice to discuss how the taxonomic resolution achieved by the ITS amplicon sequencing could be correlated to the monitoring of fungal communities associated with GTD. Specifically, ASVs of *P. chlamydospora*, often associated with grapevine trunk disease, was also detected at high relative abundance in trunk samples with metadata 'healthy' (ITS amplicon sequencing data-Table S1). How could this data be correlated with healthy/diseased status?
2. Another limitation of this study is the lack of mock fungal communities to validate the taxonomic diversity observed with different primers. This needs to be added to the discussion section.
3. Missing citation and references in the methods section for lima, DADA2, UNITE, RDP and R.
4. Primer Sequences or references for primers used in the study are missing.
5. Minor typos (missing bracket in line 288; 'pb' instead of 'bp' in line 162; numbers not in standard format: e.g. 15'000 instead of 15,000)

Staff Comments:

Preparing Revision Guidelines

Please return the manuscript within 60 days; if you cannot complete the modification within this time period, please contact me. If you do not wish to modify the manuscript and prefer to submit it to another journal, please notify me of your decision immediately so that the manuscript may be formally withdrawn from consideration by Microbiology Spectrum.

Reviewer #1 (Comments for the Author):

This study compares the differences in taxon detection and identification using different read lengths in a PacBio metabarcoding approach. Overall the study has been carefully elaborated and the manuscript is well written, and I cannot find any major flaws. However, I suggest the authors use the opportunity to add 97% OTU clustering to the methods, to enable comparison with their ASV approach when recovering taxon diversity. The ASV approach is more accurate, and so it would provide useful comparative data to have the taxonomic diversity reported for the same underlying data when doing 97% OTU clustering (overestimation?), an outdated but still widely used approach.

RESPONSE: We agree that this is a very useful alternative data point. We have now added additional analyses including a Supplementary Table S2 contrasting the ASV vs. the OTU-based taxonomy assessments. As expected, overall the changes are very minor. We mention these analyses now in a new paragraph at the end of the results.

Reviewer #2 (Comments for the Author):

The study illustrates the use of third-generation long read sequencing to characterize grapevine mycobiome. Authors test influence of different amplicon barcoding strategies for better taxonomic resolution and phylogenetic coverage. Based on the data shorter amplicon reads (full length ITS) has better taxonomic resolution and sequencing depth than longer ITS-28S amplicon reads. The manuscript is clear, concise and well-written.

There are a few minor suggestions/questions that need to be addressed:

*1. Though not the main aim of the manuscript, it would be nice to discuss how the taxonomic resolution achieved by the ITS amplicon sequencing could be correlated to the monitoring of fungal communities associated with GTD. Specifically, ASVs of *P. chlamydospora*, often associated with grapevine trunk disease, was also detected at high relative abundance in trunk samples with metadata 'healthy' (ITS amplicon sequencing data-Table S1). How could this data be correlated with healthy/diseased status?*

RESPONSE: We are hesitant to expand the discussion beyond what we have already mentioned about recovering e.g. *P. chlamydospora*. We believe that the setup of our experiment (selection of diseased/healthy individuals at random without further controls) is far from ideal to provide clear answers about the association of specific ASVs with the disease status. Hence, we prefer to keep the discussion fairly brief to avoid too speculative statements.

2. Another limitation of this study is the lack of mock fungal communities to validate the taxonomic diversity observed with different primers. This needs to be added to the discussion section.

RESPONSE: We now discuss that useful additional evaluations would indeed include the analysis of mock communities.

3. Missing citation and references in the methods section for lima, DADA2, UNITE, RDP and R.

RESPONSE : We have now included the erroneously omitted references.

4. Primer Sequences or references for primers used in the study are missing.

RESPONSE : The primer sequences are now explicitly mentioned in the methods text to avoid any misunderstanding to which published sequences we have referred to.

5. Minor typos (missing bracket in line 288; 'pb' instead of 'bp' in line 162; numbers not in standard format: e.g. 15'000 instead of 15,000)

RESPONSE: Typos fixed.

Staff Comments:

Preparing Revision Guidelines

- Point-by-point responses to the issues raised by the reviewers in a file named "Response to Reviewers," **NOT IN YOUR COVER LETTER.**
- Upload a compare copy of the manuscript (without figures) as a "Marked-Up Manuscript" file.
- Each figure must be uploaded as a separate file, and any multipanel figures must be assembled into one file.
- Manuscript: A .DOC version of the revised manuscript
- Figures: Editable, high-resolution, individual figure files are required at revision, TIFF or EPS files are preferred

Please return the manuscript within 60 days; if you cannot complete the modification within this time period, please contact me. If you do not wish to modify the manuscript and prefer to submit it to another journal, please notify me of your decision immediately so that the manuscript may be formally withdrawn from consideration by Microbiology Spectrum.

November 5, 2022

Prof. Daniel Croll
University of Neuchâtel
Emile-Argand 11
Neuchâtel
Switzerland

Re: Spectrum02513-22R1 (Quantifying trade-offs in the choice of ribosomal barcoding markers for fungal amplicon sequencing: a case study on the grapevine trunk mycobiome)

Dear Prof. Daniel Croll:

Your manuscript has been accepted, and I am forwarding it to the ASM Journals Department for publication. You will be notified when your proofs are ready to be viewed.

Sincerely,

Matthew Anderson
Editor, Microbiology Spectrum

Journals Department
Supplemental file 1: Accept